# Whole Genome Sequencing Shows that African Swine Fever Virus Genotype IX Is Still Circulating in Domestic Pigs in All Regions of Uganda

**DOI:** 10.3390/pathogens12070912

**Published:** 2023-07-06

**Authors:** Rodney Okwasiimire, Joseph F. Flint, Edrine B. Kayaga, Steven Lakin, Jim Pierce, Roger W. Barrette, Bonto Faburay, Dickson Ndoboli, John E. Ekakoro, Eddie M. Wampande, Karyn A. Havas

**Affiliations:** 1Central Diagnostic Laboratory, College of Veterinary Medicine, Animal Resources and Biosecurity, Makerere University, Kampala P.O. Box 7062, Uganda; rodn.ok17@gmail.com (R.O.); edrinekayaga2@gmail.com (E.B.K.); dickson.ndoboli@gmail.com (D.N.); eddiewampande@gmail.com (E.M.W.); 2Department of Public and Ecosystem Health, College of Veterinary Medicine, Cornell University, Ithaca, NY 14853, USA; jff22@cornell.edu (J.F.F.); jekakoro@vet.k-state.edu (J.E.E.); 3Foreign Animal Disease Diagnostic Laboratory, National Veterinary Services Laboratories, Veterinary Services, Animal and Plant Health Inspection Services, United States Department of Agriculture, Greenport, NY 11957, USA; steven.lakin@usda.gov (S.L.); jim.l.pierce@usda.gov (J.P.); roger.w.barrette@usda.gov (R.W.B.); bonto.faburay@usda.gov (B.F.)

**Keywords:** African swine fever virus, genotype, whole genome sequencing, Uganda

## Abstract

Blood samples were collected from pigs at six abattoirs in the Kampala, Uganda metropolitan area from May 2021 through June 2022, and tested for African swine fever virus. Thirty-one samples with cycle threshold values < 26 from pigs with different geographic origins, clinical and pathologic signs, and *Ornithodoros moubata* exposure underwent whole genome sequencing. The p72 gene was used to genotype the isolates, and all were found to be genotype IX; whole genome sequences to previous genotype IX isolates confirmed their similarity. Six of the isolates had enough coverage to evaluate single nucleotide polymorphisms (SNPs). Five of the isolates differed from historic regional isolates, but had similar SNPs to one another, and the sixth isolate also differed from historic regional isolates, but also differed from the other five isolates, even though they are all genotype IX. Whole genome sequencing data provide additional detail on viral evolution that can be useful for molecular epidemiology, and understanding the impact of changes in genes to disease phenotypes, and may be needed for vaccine targeting should a commercial vaccine become available. More sequencing of African swine fever virus isolates is needed in Uganda to understand how and when the virus is changing.

## 1. Introduction

The African swine fever virus (ASFV) is a double-stranded DNA arbovirus with a genome size of 170 to 190 kb, which encodes over 150 proteins, depending on the viral strain [1], and belongs to genus *Asfivirus*, family *Asfarviridae* [2]. ASFV causes an infectious disease called African swine fever (ASF), which is a highly contagious hemorrhagic disease that has been reported in both domestic and wild suids including warthogs, bush pigs, giant forest hogs, and wild boars [3]. Domestic pigs are highly susceptible, and case fatality rates can approach 100% [4]. Though the virus does not infect humans, ASF has led to food insecurity and enormous economic losses to farmers due to the high mortality rates of pigs during outbreaks [5,6] and disruptions of the market structure in the pig value chain [7].

The ASFV was first reported in Kenya in the early 20th century [8]. There have been 24 genotypes of the virus reported based on genome sequencing of the p72 segment of the virus since this initial discovery [9]. It is evident that ASFV variants can quickly spread across regions. Currently, there is a genotype II variant causing a global panzootic that was first diagnosed outside of Africa in the country of Georgia [10] and has spread widely since that introduction, impacting Europe, Asia, and the island of Hispaniola (www.wahis.woah.org, accessed on 27 February 2023). Given this background, the risk of introduction of new ASFV genotypes into Uganda is likely to occur, and ongoing monitoring is needed.

In Uganda, sequencing of the p72 gene of the virus from ASF outbreaks in Central Uganda in 2007 [11], between 2010 and 2013 from all districts in Uganda [12], and 2015 from the Central [13] and Eastern region [5] detected only genotype IX. An ASFV genotype X virus was found from a Ugandan pig in isolates that were part of the ASFV collection at the Institute for Animal Health in the Pirbright Laboratory [14,15]. The neighboring country of Kenya reported genotype IX in warthogs and X in domestic pigs [16]. Another neighbor, Tanzania, has multiple genotypes in domestic pigs. Isolates from pigs in Tanzania were sequenced and genotypes X, XV, and XVI were found [17], an outbreak in 2008 was caused by an ASFV genotype XV variant [18], and samples collected between 2015 and 2017 in Tanzania were found to be infected with ASFV genotypes II, IX, and X [19]. Sequenced samples from domestic pigs in the Democratic Republic of the Congo (DRC), a country on Uganda’s western border, reported detection of genotypes IX [20], and genotype X [21]. Further, sequencing of samples collected from domestic pigs between 2005 and 2012 detected genotypes I, IX, and X in the DRC [22] Numerous genotypes are commonly reported among Uganda’s neighbors. Given the diversity of genotypes in the region, it is important to track their presence in the country to understand when new genotypes emerge and why.

Previous work in Uganda has been constrained to outbreak locations and to collections of samples at global reference laboratories. Genotyping of isolates that were representatively collected and that were associated with a variety of characteristics (clinical signs, region, tick exposure, etc.) had not been done. Further, maintenance of knowledge about circulating strains is important for the development of appropriate vaccine candidates for a given region/country and for purposes of molecular epidemiology. The purpose of this study was to determine what genotypes were circulating based on samples of varying characteristics, and to establish whether any new ASFV genotypes were circulating in Uganda.

## 2. Materials and Methods

### 2.1. Sample Collection

A total of 1318 pigs had blood samples collected at six abattoirs around the Kampala metropolitan area in Central Uganda (Figure 1). In addition, pig sex, type (local breed, European breed, or cross-bred), clinical signs, and pathologic lesions were recorded at the time of sample collection. Traders were asked about the origin district of the pig as well. Abattoirs in metropolitan Kampala have been reported to receive pigs from all the regions of the country [23]. Uganda has four geographic regions, including the Central, Northern, Western and the Eastern (Figure 1). A stratified systematic sampling method was followed from May 2021 through June 2022. Total sample sizes were calculated to capture approximately 200 positive pigs. To detect the expected ASFV prevalence of pigs in Uganda, 11.5% [24], with 95% confidence and 5% error, 157 pigs would be needed (openepi.com; accessed July 2018). This would yield 18 positive pigs. To detect 200 positive pigs, at least 1200 pigs were sampled. The total sample size was stratified across abattoirs based on the annual number of pigs slaughtered at each site. This was then calculated to monthly sample sizes. Sites had two to four days per month randomly selected for sampling so as to not exceed 10 pigs sampled per day to allow enough time to collect samples from all of the pigs. On the day of sampling, pigs were sampled systematically until the sample size was met. The sampling team consisted of trained veterinarians to ensure proper capture of clinical signs and pathologic lesions, as well as appropriate sample collection. Blood samples were transported from the slaughterhouse to the Central Diagnostic Laboratory, College of Veterinary Medicine, Animal Resources and Biosecurity, Makerere University using a cold chain where they were stored at −20 °C.

### 2.2. Nucleic Acid Extraction, Amplification, and Detection of ASFV by qPCR

Whole blood was diluted 1:1 with 1X PBS and total DNA extraction was performed using the Qiagen DNeasy tissue and blood kits (Qiagen, Hilden, Germany). The US Department of Agriculture’s (USDA) Foreign Animal Disease Diagnostic Laboratory’s (FADDL) standard operating procedures (SOPs) [25] which follow the manufacturer’s instructions were used during these extractions. The real-time PCR (qPCR) assay used has been previously described [26] and the FADDL ASF qPCR SOP [27] was again followed. The TaqMan^®^ Fast Virus 1-Step Master Mix (Thermo Fisher Scientific, Waltham, MA, USA) along with the forward primer of 5′-CCTCGGCGAGCGCTTTATCAC-3′, reverse primer of 5′-GGAAACTCATTCACCAAATCCTT-3′, and probe of FAM-CGATGCAAGCTTTAT-MGB/NFQ ordered from Thermo Fisher Scientific (Waltham, MA, USA) were used in the qPCR procedure. The VetMax Xeno DNA internal positive control (IPC) (Thermo Fisher Scientific, Waltham, MA, USA) was used during the DNA extraction procedures and the VetMax Xeno IPC LIZ Assay (Thermo Fisher Scientific, Waltham, MA, USA) was used during the real time qPCR. This was done for each individual sample following FADDL SOPs. The qPCR assay was run on a QuantStudio 5 thermocycler (Thermo Fisher Scientific, Waltham, MA, USA) in 25 µL reaction volumes containing 20 µL of master mix and 5 µL of the extracted total DNA.

### 2.3. Blood Sample Selection

Blood samples used for sequencing were positive based on qPCR results and had a cycle threshold (Ct) value < 26. In total, 31 qPCR positive samples were sequenced. Samples from different regions of the country (See Figure 1), and different districts (for the central region), as well as those from pigs that had presented with and without clinical signs and pathologic lesions at the time of sampling at the abattoir were considered for selection. The samples sequenced also covered a range of *O. moubata* exposure status of the pigs (See Table 1). Table 1 also summarizes the distribution of pig sex and type by region, although they were not used for sample selection. It is important to note that the intent was to sequence a diverse set of positive samples, but not necessarily a representative set of samples as there is no national level data on disease prevalence across Uganda.

### 2.4. African Swine Fever Genome Sequencing

Following sample selection, the extracted DNA previously used for qPCR testing was prepared for sequencing with minor modifications from the protocol previously described [28], adapted from the Nextera XT DNA Sample Preparation Guide (Illumina, San Diego, CA, USA, 2019) [29]. All the reagents used were supplied with the Nextera XT DNA Library Prep sequencing kit (Illumina, San Diego, CA, USA) unless otherwise stated. Briefly, the DNA was quantified on a Qubit 4 fluorometer and 1.0 ng from each DNA sample was fragmented and adapter sequences added to the ends to allow amplification by limited-cycle PCR in downstream steps. The incubation time for the fragmentation and tagmentation was increased to 15 min to allow for generation of DNA fragments of appropriate sequencing size. The sizes of the fragments produced were examined by capillary electrophoresis on a 5200 Fragment Analyzer System (Agilent, Santa Clara, CA, USA). For the PCR amplification, Nextera PCR Master Mix (NPM) Index 1 (i7) and 2 (i5) primers in a TruSeq index plate fixture were utilized. The PCR was carried out in a 96-well plate on a SimpliAmp™ thermocycler (Thermo Fisher Scientific, Waltham, MA, USA) following the limited PCR program outlined in the Nextera XT DNA Sample Preparation Guide. The sizes of the fragments produced were examined by capillary electrophoresis on a 5200 Fragment Analyzer System (Agilent, Santa Clara, CA, USA). The PCR products were cleaned using AMPure XP beads (Beckman Coulter, Indianapolis, IN, USA), washed with freshly prepared 80% ethanol on a magnetic stand and suspended into 50 µL resuspension buffer (RSB) supplied in the sequencing kit.

The resultant libraries were pooled in equal concentrations to create a pooled amplicon library (PAL) of 4 nM. The PAL was denatured according to manufacturer’s instructions to create a diluted amplicon library (DAL) of 14 pM. The DAL was loaded into a thawed MiSeq V3 600 cycle reagent cartridge for sequencing on the Illumina MiSeq platform (Illumina, Inc., San Diego, CA, USA). Sequencing occurred at the Makerere University College of Health Science Biomedical Research Centre (MakBRC, Kampala, Uganda) Sequencing Laboratory.

### 2.5. Sequence Data Analysis

Samples were analyzed using methods and software as previously described by Lakin et. al. [30]. Briefly, Illumina data were aligned to the ASFV Kenya Bus/2006 reference genome (GenBank accession KM111295.1) using the Burrows-Wheeler Aligner (v0.7.17) with options “-a-h 2-Y-M” (Li & Durbin, 2012). Freebayes parallel (v1.3.4) was used to call insertions and deletions for the Illumina data with the option “standard-filters” (Garrison and Marth, 2012). The publicly available vSNP pipeline (USDA, https://github.com/lakinsm/simple-snp, accessed on 27 December 2022) was used to visualize SNPs for the epidemiological analysis calculated using an open-source SNP caller (https://github.com/lakinsm/simple-snp, accessed on 27 December 2022). Variants were filtered to meet the following thresholds: a minimum depth of 10 observed alleles at a genomic location across the population of samples (DP > 10), a minimum observed alternate allele count of 7 at a given genomic location across the population of samples (AO > 7), and an alternative allele frequency greater than or equal to 70% at a given site within a given sample.

To construct the phylogenetic tree, a total of 46 ASFV genomes from public databases were aligned alongside the Illumina sequencing data using the Burrows-Wheeler Aligner. Reference genomes for Genotype IX (ASFV Kenya Bus/2006 KM111295.1) and X (ASFV Kenya Tk1/2005 NC_044945.1 and ASFV Kenya/1950 NC_044944.1) were used during alignment, and SNPs were called as described above. A SNP-based phylogenetic tree was generated using RAxML (v8.2.12).

### 2.6. Mapping

QGIS version 3.28.1 Firenze (qgis.org) was used for mapping. Uganda district shp files were downloaded from the United Nations (UN) Human High Commissioner for Refugees (https://data.unhcr.org/en/documents/details/83043, accessed on 30 March 2023) and regional data was downloaded from the Office for the Coordination of Humanitarian Affairs (https://data.humdata.org/dataset/cod-ab-uga, accessed on 29 June 2023). Districts were linked to data on the number of samples sequenced per district and mapped.

## 3. Results

### 3.1. Genotype Characterization

This study had sequencing results for 31 blood samples that were positive for ASFV with Ct values of less than 26. The pigs from which the blood samples were taken vary by region and district of origin, clinical presentation, and *O. moubata* tick exposure. The samples were taken from a larger set of 1318 blood samples representatively collected between May 2021 and June 2022. All 31 ASFV isolates were classified as genotype IX based on their p72 sequences and whole genome similarity to known genotype IX sequences.

### 3.2. Sequence Analysis

Of the 31 ASFV samples sequenced, six had sufficient depth of coverage across the genome (>7x) to characterize variants. Although these six isolates aligned closely to a Genotype IX virus found in Uganda in 2015, they established their own clade, suggesting further evolution of the virus (Figure 2). Further, one of these six new isolates (S24/2021) differed from the other five isolates.

All six ASFV genotype IX sequences characterized in this study diverged from Kenya genotype X sequences (Kenya/1950, Tk1/2005) by over 2500 SNPs (>1%) of the genome and grouped with historic genotype IX sequences from Kenya (Bus/2006) and Uganda (2015 isolates). The newly characterized genotype IX sequences shared approximately 100 single nucleotide polymorphisms (SNPs) with the historic genotype IX sequences but diverged from the 2015 Ugandan sequences by approximately 20–60 SNPs not previously described in the 2015 sequences. Five of the six isolates characterized in this study were closely related and clustered into the same clade, while one sequence (S24/2021) appeared to be more ancestrally related to the previously sequenced Ugandan isolates. The SNPs characterized occurred throughout the genome and were not isolated to any region, gene, or multigene family. Further sequencing is needed to thoroughly describe these isolates and to elucidate the level of difference, but the results did show that there were SNPs that were shared among five of this study’s isolates, and the sixth isolate also had a unique SNP fingerprint.

## 4. Discussion

This study sequenced 31 ASFV detected in blood samples collected between May 2021 through June 2022 from pig abattoirs in the Kampala metropolitan area of Uganda. All samples were identified as genotype IX based on p72 sequence analysis. This aligns with previous work done on in 2007 [11], 2013 [12], and 2015 [13,31]. There have also been two isolates classified as genotype X from Uganda, one from 1965 and another from 1995 [14]. Although we cannot definitively say that other genotypes are not present in Uganda, it appears as though genotype IX is a stable and common cause of ASFV despite the fact that neighboring countries that have a shared border with Uganda have various other genotypes circulating [16,17,18,19,20,21,22,32]. Given that the genotype II that was introduced into the Republic of Georgia [10] has spread globally in the same period of time, this suggests that there is not a rapid regional movement of variants, although studies that would target sequencing of isolates in high-risk areas of entry would better determine if there were any incursions of new genotypes.

There were six ASFV sequences that had enough depth of coverage to further evaluate. It was found that they created their own clade and one of the six differed from the other five variants and had its own SNP pattern (Appendix A). The differences detected among these isolates suggests that viruses continue to evolve within genotypes and specific geographical locations, as has been previously described for the genotype II epizootic ongoing since 2008 [33]. The impact of this evolution on considerations such as clinical presentation and pathologic presentation, as well as transmission efficacy from *O. moubata* or between pigs will require further study.

In this study the ASFV genotype IX was confirmed to be circulating widely in Uganda, but the work also revealed that the viruses in this genotype continue to evolve, creating diversity within the genotype and the country. There is a need for researchers to leverage whole genome sequencing and to develop a more robust database of African swine fever sequences for comparison to track this evolution and its impact. The p72 gene segment has allowed for genotyping [14] of ASFV, but with whole genome sequencing technology, more detailed comparisons of viruses and their evolution are possible. Such work will allow better understanding of relationships between the genome and disease presentation and is critical to fully leverage molecular epidemiology in outbreak responses, which can determine transmission dynamics and spatiotemporal trends.

## Figures and Tables

**Figure 1 pathogens-12-00912-f001:**
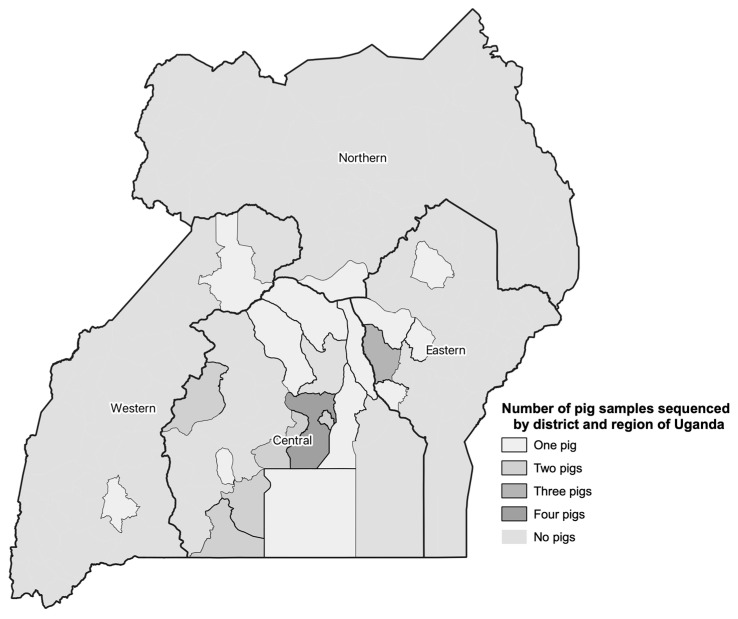
Summary of the number of pig blood samples collected from May 2021 through June 2022 sequenced by administrative district and region in Uganda.

**Figure 2 pathogens-12-00912-f002:**
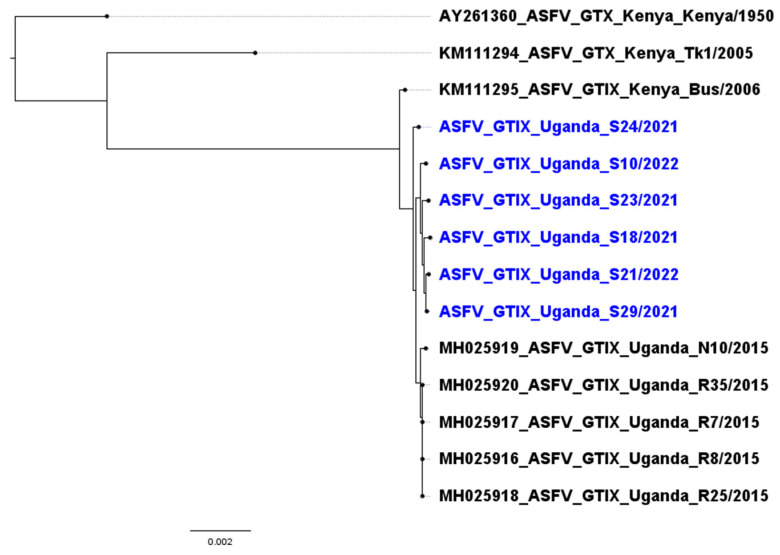
Phylogenetic tree of six isolates sampled from Uganda in 2021–2022 (blue) compared to historic genotype X isolates from Kenya and genotype IX isolates from Kenya and Uganda.

**Table 1 pathogens-12-00912-t001:** Characteristics of the 31 blood samples collected from May 2021 through June 2022 that were selected for sequencing to determine the African swine fever virus genotype.

*n* = 31	Central	Eastern	Northern	Western
	# (%)	# (%)	# (%)	# (%)
Clinical signs				
Yes	8 (25.8)	2 (6.5)	1 (3.2)	0 (0.0)
No	12 (38.7)	6 (19.4)	0 (0.0)	2 (6.5)
Pathologic lesions				
Yes	16 (51.6)	8 (25.8)	1 (3.2)	2 (6.5)
No	4 (12.9)	0 (0.0)	0	0
Ornithodoros				
Negative	7 (22.6)	3 (9.7)	0 (0.0)	0 (0.0)
Weak Positive	9 (29.0)	3 (9.7)	1 (3.2)	1 (3.2)
Positive	1 (3.2)	0 (0.0)	0 (0.0)	0 (0.0)
Strong Positive	3 (9.7)	2 (6.5)	0 (0.0)	0 (0.0)
Sex				
Male	6 (19.35	3 (9.7)	1 (3.2)	2 (6.5)
Female	14 (45.2)	5 (16.1)	0 (0.0)	0 (0.0)
Pig type				
Local	1 (3.2)	3 (9.7)	0 (0.0)	0 (0.0)
European	9 (29.0)	3 (9.7)	1 (3.2)	0 (0.0)
Cross-bred	9 (29.0)	2 (6.5)	0 (0.0)	2 (6.5)

Clinical signs, pathologic lesions, *Ornithodoros moubata* exposure and region were used for sample selection.

## Data Availability

Sequences may be available upon request with the permission of the Ugandan Ministry of Agriculture, Animal Industries and Fisheries.

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
