# Peer review of "Whole Genome Sequencing Shows that African Swine Fever Virus Genotype IX Is Still Circulating in Domestic Pigs in All Regions of Uganda"

_pathogens, 2023, doi:10.3390/pathogens12070912_

Round 1

Reviewer 1 Report

Article entitled “Whole genome sequencing shows that African swine fever virus geno-2 type IX is still circulating in domestic pigs in all regions of Uganda.” has scientific value. 

Comments

A quality article providing important and interesting information about the epidemiology of the ASFV.

Performed on a good scientific basis with adequate methods, the results obtained provide important scientific information.

However, there are a number of gaps in the article.

The article provides a whole genome sequencing of the ASF virus, however, access to the data obtained is limited. This makes it impossible for other scientists to analyze the data. The authors also report an extremely high level of SNIP in sequenced genomes (about 2500). This is indeed a very large number. However, the authors do not analyze either the distribution of SNIPs in the virus genome or the reasons for such a large number of SNIPs. It is possible to supplement the relevant information in supplementary materials where there is a partial but insufficient analysis of mutations.

The authors indicate that the virus is "circulating in domestic pigs in all regions of Uganda" however, from the map (Figure 1) it appears that the virus is circulating in a relatively limited region of the country.

The map shown in the article is too small and difficult to perceive. Perhaps it should be replaced by a map with the designation of detected cases of ASF.

Reviewer 2 Report

Rodney Okwasiimire et al presented a research finding entitled on 'Whole genome sequencing showing that African swine fever virus genotype IX is still circulating in domestic pigs in all regions of Uganda'. Overall, the strength of the paper is its  study methodology, and discussions. TIt is well designed and written in good order. However, the authors need to consider to address the following limitations.

1/ They claimed stratified systematic sampling had been used to sample the animals for blood screening and whole genome sequencing. This lacks clarity how the sampling strategy is executed. 

2/ Details regarding animals data such as age, sex, and history are missed on the sampling strategies and methodologies.

3/ The study seems molecular epidemiological investigation of the ASV virus in all regions of Uganda, however, the result didn't clearly reflect the data how the circulating virus is prevalent on the basis of different epidemiological parameters.

4/ It would be great if authors describe in which facility the sequencing service was given or performed.

5/ Sequencing data should be provided as a supplementary in the publication 

6/ Authors didn't give much attention when proofreading their paper. some contents are copied and paste from other resources. This can be a sign of plagiarism practice and should be avoided. For example:On Author contributions part,  the first two statements are copied as it is from journal's author instruction on how to write individual contributions.  

Even on the body of the text,  there is detectable degree of plagiarism and own words and statements should be used. 

7/Line 48 to 50, it is described that the likelihood of the ASV virus to Uganda. What is the evidence here? 

8/ Keywords shouldn't  be written with numbers. Careful revision is required. 

Needs improvement.

Round 2

Reviewer 2 Report

I am satisfied that authors have addressed comments and suggestions properly.